# Rapid Discrimination and Prediction of Ginsengs from Three Origins Based on UHPLC-Q-TOF-MS Combined with SVM

**DOI:** 10.3390/molecules27134225

**Published:** 2022-06-30

**Authors:** Chi Zhang, Zhe Liu, Shaoming Lu, Liujun Xiao, Qianqian Xue, Hongli Jin, Jiapan Gan, Xiaonong Li, Yanfang Liu, Xinmiao Liang

**Affiliations:** 1CAS Key Laboratory of Separation Science for Analytical Chemistry, Dalian Institute of Chemical Physics, Chinese Academy of Sciences, Dalian 116023, China; zhangchi@jcmsc.cn (C.Z.); liuzhe@dicp.ac.cn (Z.L.); lushaoming@jcmsc.cn (S.L.); xiaoliujun@jcmsc.cn (L.X.); ganjiapan@jcmsc.cn (J.G.); liuyanfang@jcmsc.cn (Y.L.); liangxm@dicp.ac.cn (X.L.); 2Jiangxi Provincial Key Laboratory for Pharmacodynamic Material Basis of Traditional Chinese Medicine, Ganjiang Chinese Medicine Innovation Center, Nanchang 330000, China; lixiaonong@jcmsc.cn; 3University of Chinese Academy of Sciences, Beijing 100049, China

**Keywords:** ginseng, UHPLC-Q-TOF-MS, discrimination, support vector machine

## Abstract

Ginseng, which contains abundant ginsenosides, grows mainly in the Jilin, Liaoning, and Heilongjiang in China. It has been reported that the quality and traits of ginsengs from different origins were greatly different. To date, the accurate prediction of the origins of ginseng samples is still a challenge. Here, we integrated ultra-high-performance liquid chromatography quadrupole time-of-flight mass spectrometry (UHPLC-Q-TOF-MS) with a support vector machine (SVM) for rapid discrimination and prediction of ginseng from the three main regions where it is cultivated in China. Firstly, we develop a stable and reliable UHPLC-Q-TOF-MS method to obtain robust information for 31 batches of ginseng samples after reasonable optimization. Subsequently, a rapid pre-processing method was established for the rapid screening and identification of 69 characteristic ginsenosides in 31 batches ginseng samples from three different origins. The SVM model successfully distinguished ginseng origin, and the accuracy of SVM model was improved from 83% to 100% by optimizing the normalization method. Six crucial quality markers for different origins of ginseng were screened using a permutation importance algorithm in the SVM model. In addition, in order to validate the method, eight batches of test samples were used to predict the regions of cultivation of ginseng using the SVM model based on the six selected quality markers. As a result, the proposed strategy was suitable for the discrimination and prediction of the origin of ginseng samples.

## 1. Introduction

Ginseng is the dried root of *Panax ginseng* C. A. Mey, first recorded in the *Shennong’s Classic of Materia Medica*. It has been widely used in many disease for more than two thousand years, because of its wide range of pharmacological effects [1]. Modern pharmacological studies have shown that ginseng has various pharmacological activities such as anti-tumor [2], anti-oxidative [3], improving immunity [4], and enhancing memory [5]. In China, ginseng is mainly planted in the northeast regions, including Jilin (JL), Liaoning (LN), and Heilongjiang (HLJ). According to reports, the quality and traits of ginseng from different origins shows great diversity, due to different cultivation techniques and ecological environments [6,7]. Therefore, it is imperative to establish a method of quality evaluation to differentiate and characterize ginseng samples from different regions.

Phytochemical studies have revealed the major compositions in ginseng, including ginsenosides, polysaccharides, amino acids, polypeptides proteins, and volatile oils [8]. Among them, ginsenosides are considered the main active components [9,10,11,12,13]. In the 2020 edition of the *Pharmacopoeia of the People’s Republic of China* (ChP), only three ginsenosides and their contents were used as standards for quality evaluation of ginseng, making it impossible to distinguish ginsengs from different origins [14]. In recent years, methods based on liquid chromatography mass spectrometry (LC-MS) fingerprint, LC-MS quantification, and chemical pattern recognition have been widely used to solve this issue [15,16,17]. Xiu et al. quantified fourteen ginsenosides using UHPLC coupled with triple quadrupole mass spectrometer (QQQ-MS). Two commonly used traditional multivariate statistical analysis methods, principal component analysis (PCA) and partial least squares discriminant analysis (PLS-DA) were further employed to evaluate differences in the contents of these ginsenosides between origins [18]. However, these methods still lacked objectivity and accuracy in their identification results. Additionally, the established QQQ quantitative method required fourteen reference standards for content determination, resulting in a high detection cost and poor practicality of this method. Thus, it is essential to develop a convenient, effective strategy for the accurate differentiation and characterization of ginsengs from different regions of cultivation.

Recently, the combination of UHPLC-MS and support vector machines (SVM) has been considered as a valid method for the authentication of species and the identification of origins for Traditional Chinese Medicines (TCMs), with satisfactory accuracy [19,20]. For instance, Zhao et al. [19] managed to distinguish different varieties of ginsengs using UHPLC-MS integrated with SVM and accurately distinguished the red ginseng from other ginseng samples (white ginseng, *Panax quinquefolium*, and *Panax notoginseng*) after sufficient training. However, ginseng from different origins exhibited high similarity in chemical composition, which increased the difficulty of identification. Thus, higher requirements in establishing a model and data processing of SVM is required. In addition, as far as we know, the discovery of quality markers based on SVM model remains challenging.

In this work, a rapid, convenient, and effective differentiation method based on UHPLC-Q-TOF-MS coupled with SVM was developed to evaluate ginseng samples collected from JL, LN, and HLJ. Firstly, stable and reliable data were generated by UHPLC-Q-TOF-MS, and common ginsenosides components of 31 batches of ginsengs were screened. Additionally, the SVM model was established to accurately classify ginseng from different origins using the normalized data. Furthermore, an algorithm of feature contribution values was introduced to the SVM model to obtain quality markers of the ginsengs from three origins. Finally, on the basis of these quality markers, the SVM model was shown to be able to discriminate and predict the geographical origins of ginseng. This strategy was verified by successfully distinguishing test samples from JL, LN, and HLJ, indicating great reliability and affectivity. Our strategy has the potential to provide references for the regional differentiation and traceability of other TCMs.

## 2. Results and Discussion

### 2.1. Development of Analysis Method of Ginsengs from Different Origins 

#### 2.1.1. Optimization of UHPLC-Q-TOF-MS Analysis Conditions

In order to achieve good separation effects and obtain high-quality UHPLC-Q-TOF-MS data, we optimized the extraction method, extraction solvents, composition of mobile phase, elution gradient, and injection concentration in detail during UHPLC-Q-TOF-MS analysis. The results showed that 40% ethanol with ultrasonic is suitable for the extraction of ginseng and that water (containing 0.01% formic acid, *v*:*v*) and acetonitrile (containing 0.01% formic acid, *v*:*v*) are preferred by the mobile phase system due to higher peak numbers and better resolution. The injection concentration of 20,000 ppm can obtain an excellent response and will not burden the instrument. These results are shown in Appendix A. Those results show that the optimization of UHPLC-Q-TOF-MS analysis conditions when used for ginseng is necessary to ensure that the samples enter the subsequent analysis in the best state.

#### 2.1.2. Validation of the UHPLC-Q-TOF-MS Analysis Method

After the development of UHPLC-MS analysis conditions, the method was verified by QC samples. The stability and repeatability of the system were evaluated by extraction ion chromatograms (EICs) in QC samples. QC samples were run before and after injection every day, and one QC was inserted every ten samples during the injection. As shown in Appendix A, information for a total of seven EICs was extracted from QC. The mass accuracy RSDs of those seven EICs was calculated to be from 1.10 × 10^−4^% to 1.46 × 10^−4^%, the RSDs of the retention time were from 0.06% to 0.43%, and the RSDs of the peak area were from 1.94% to 2.43%. The results showed good stability and repeatability of UHPLC-Q-TOF-MS. The analytical environment constructed by UHPLC-Q-TOF-MS can meet the needs of sample analysis and obtain real and robust data.

### 2.2. Rapid Screening and Identification of Characteristic Ginsenosides in Ginsengs from Different Regions 

The developed UHPLC-Q-TOF-MS method was subsequently applied to the analysis of 31 batches of ginseng samples from JL, LN, and HLJ, and MS data were collected. The total ion chromatogram (TIC) of the ginseng sample by UHPLC-Q-TOF-MS is shown in Appendix A. We established a pre-processing method to rapidly filter high-quality information from redundant mass data for data analysis. 

According to the processing of the workstation, we screened more than 6000 pieces of information from 31 batches of samples. After peak matching, alignment, and filtering, 122 common peaks were found in the data, and these common peaks were present in all 31 batches of ginsengs. Furthermore, we made comparisons using the in-house database (including MS and MS/MS information of over 400 ginsenosides collected from published references); 69 ginsenosides were quickly screened (Figure 1), and their chemical structures were preliminarily identified (Table 1).

Based on the in-house database, the fragmentation patterns of three typical ginsenosides were summarized. In PPT-type ginsenosides, such as Rg1, the parent ion [M-H]^-^(*m*/*z* 799) in the negative-ion mode showed a loss of two glucose to obtain aglycone protopanaxatriol(*m*/*z* 475), as shown in Appendix A. In PPD-type ginsenosides, such as Rb1, the parent ion [M-H]^-^(*m*/*z*) in negative-ion mode showed a loss of four glucose residues to obtain protopanaxadiol(*m*/*z* 459), as shown in Appendix A. In OA-type ginsenosides, such as Ro, the parent ion [M-H]^-^(*m*/*z* 955) in negative-ion mode showed a loss of two glucose residues and one glucuronic acid group to produce oleanolic acid (*m*/*z* 455), as shown in Appendix A. In the negative-ion mode, the parent ion information was obtained by full scanning of UHPLC-Q-TOF-MS, also known as the MS1 fragment, which mainly exists in the form of [M-H]^-^ and [M+HCOO]^-^. These were common adduct ion forms of ginsenoside, which is also consistent with the literature [16]. Under MS/MS mode, the sugar on the branched chain gradually cracked, and finally, a relatively stable parent nucleus with *m*/*z* of 475, 459, and 455 was detected in three typical ginsenoside styles [21]. In addition, it was found that these parent nucleus were not easy to cleave, and this fragment information is an important basis for our identification and classification of unknown ginsenosides. The full scan mode and MS/MS mode of six compounds, including Compound 13(ginsenoside Rg1), Compound 14(ginsenoside Re), Compound 27(ginsenoside Rf), Compound 46(ginsenoside Rb1), Compound 50(ginsenoside Rc), and Compound 56(ginsenoside Rb2) are shown as examples in Appendix A. These experimental results are consistent with the rules obtained in our summary [22].

Accordingly, 69 ginsenosides were quickly screened (Figure 1), and their chemical structures were preliminarily identified (Table 1). Although the clear structure could not be determined, it does not affect the types of unknown components, nor does it affect the subsequent model analysis.

In brief, the rapid pre-processing method was used for rapid screening and identification of 69 characteristic ginsenosides in 31 batches of ginseng samples from three different origins, and the data pre-processing was performed within an hour, which provided high-quality data for the subsequent multivariate statistical analysis. 

### 2.3. Classification and Prediction of Ginsengs from Different Origins by Multivariate Statistical Analysis

#### 2.3.1. Traditional Multivariate Statistical Analysis 

Traditional multivariate statistical analyses, such as PCA and PLS-DA, were conducted using the peak areas of the 69 characteristic ginsenosides to elucidate the similarities and differences between ginsengs from three different geographical origins. 

PCA, a commonly used unsupervised data processing model, was used to discover the trends of the ginseng samples from different growing origins. The first two principal components only accounted for 33.0% of the variation. As shown in Figure 2, the 31 batches of ginseng samples failed to establish origins.

Subsequently, a supervised data model PLS-DA was established to further to identify the samples by origins. The R^2^Y and Q^2^ of PLS-DA were 0.87 and 0.56, respectively. Furthermore, the PLS-DA model was evaluated using a permutation test shown in Appendix A. In the random permutation test (Appendix A), intercepts of R^2^ and Q^2^ were 0.371 and 0.277, respectively. As shown in the PLS-DA score plot (Figure 3), the ginseng samples in the three different geographical areas were divided into only two clusters (from or not from LN), suggesting the failure of identification. This was possibly because the least-squares method cannot effectively handle nonlinear MS data.

#### 2.3.2. SVM Analysis 

As a widely used method, SVM has been successfully applied in the quality control of TCM with satisfactory classification and prediction accuracy [20]. In this work, an SVM model was developed to discriminate and predict the ginsengs from cultivation regions, using the peak areas and normalized data of 69 characteristic ginsenosides as input vectors and regions as outputs.

The best values for parameter C and parameter γ of the SVM model were calculated using a grid search method combined with ten-fold cross-validation. The parameter C affected the distance between the support vector and the decision plane. The parameter γ was mainly used to map the height of the low-dimensional samples. Classification accuracy under different combinations for γ and C are shown in Figure 4. There was a large plateau, indicating that the SVM model was well-establishment, and a γ value of 0.03 and a C value of 1 were chosen in the ten-fold cross-validation for all data.

As shown in Table 2, the 31 batches of ginseng samples using peak areas were assigned to individual origins by peak areas with a prediction accuracy of 83%. However, the accuracy of the classification of regions reached 100% when normalized data were used. Therefore, data normalization significantly improved the SVM performance because the Z-Score normalization converted each feature into a standard normal distribution. This prevented the average and variance of the features from affecting the dimensionality reduction results.

Thus, the results strongly indicated that the developed SVM model with normalized data was a powerful tool for the geographical classification and prediction of ginsengs from JL, LN, and HLJ. 

### 2.4. Discovery of Quality Markers of Ginsengs from Three Different Origins 

As far as we know, key feature extraction for SVM is still a challenge, which cannot be handled by traditional statistical methods, such as the *t*-test. To deal with this problem, a permutation importance algorithm was employed in this study. According to the formula (A9), the contribution of all peaks to the SVM was calculated. In the next step, the potential quality markers of ginsengs from JL, LN, and HLJ were selected due to the calculations. Based on the importance value (IV > 0), six quality markers were discovered, including peak 65 (AcO-ginsenoside Rd or isomer), peak 18 (AcO-ginsenoside Re or isomer), peak 26 (Ginsenoside Re2 or isomer), peak 25 (Notoginsenoside M or isomer), peak 3 (Ginsenoside Re2 or its isomer), and peak 33 (Yesanchinoside J or isomer). Their contributions were ranked from highest to lowest as shown in Figure 5. Box plots of the six quality markers are shown in Appendix A, which indicates that there were distributional differences between the same characteristics in JL, LN, and HLJ.

To prove the capability of the six quality markers, SVM model was established again using six quality markers and ten-fold cross-validation. The origin identification accuracy of ginsengs was 100%. The results of the identification of ginseng origin by SVM with six quality markers are shown in Appendix A, which indicates that the six quality markers were sufficient to identify the origin of ginseng samples. The selection of six peaks from 69 peaks simplified the process of ginseng sample data acquisition.

### 2.5. Verification of This Strategy for Ginseng Identification from Different Origins Using Test Samples

To verify the real application capability of this strategy, eight batches (T1-T8) of test ginseng samples, purchased in the market from different growth origins, were used for prediction experiments. According to the sample preparation method, analysis method, and pre-processing method of this strategy described above, normalized data of six differential markers in eight batches of ginseng samples were screened and imported into the SVM model as vectors to distinguish. As shown in Table 3, ginseng samples from three provinces were all correctly identified with an accuracy of 100%, indicating that this approach can effectively and accurately predict the geographical origin of ginseng samples sold in the market. 

## 3. Materials and Methods

### 3.1. Ginseng Samples

Ginseng samples were collected in three provinces in Northeastern China, including JL, HLJ, and LN. All samples were identified as dry roots of *Panax ginseng* CA Mey. by Xiaoping Yang from Dalian Institute of Chemical Physics, Chinese Academy of Sciences. Sample information is shown in Table 4. S1~S31 are training samples and T1-T8 are the test samples.

### 3.2. Chemicals and Reagents

LC-MS-grade acetonitrile was purchased from Fisher Scientific (Pittsburgh, PA, USA), LC-grade formic acid was purchased from Sigma, ultrapure water was obtained from Milli-Q IQ 7000 system (Bedford, MA, USA), and analytical-grade ethanol was purchased from Energy Chemical (Shanghai, China).

### 3.3. Preparation of Samples

One gram of dry ginseng powder was extracted with 50 mL of 40% ethanol using an ultrasonic method (Kunshan ultrasonic instruments Co., Ltd., Suzhou, China) for 45 min, and the extracted solution was centrifuged at 10,000 rpm for 10 min to obtain the sample stock solutions for UHPLC-Q-TOF-MS. One milliliter of solution was collected from each sample stock solution from 39 batches of ginseng and mixed to obtain Quality Control (QC) samples. All stock solutions were filtered through a 0.22 μm membrane filter prior to UHPLC-Q-TOF-MS analysis.

### 3.4. UHPLC-Q-TOF-MS Analysis of Ginseng Samples

Chromatographic separation of ginseng samples was performed on an Agilent 1290 Infinity II UHPLC system (Agilent Technologies Inc., Santa Clara, CA, USA) using an Acquity UPLC BEH C18 (2.1 × 100 mm, 1.7 µm) column (Waters Corporation, Milford, MA, USA). Mobile phases were 0.1% formic acid water (*v*:*v*, phase A) and acetonitrile (phase B), the flow rate was 0.4 mL/min, the injection volume was 1 µL, the column temperature was 30 ℃, and the detection wavelength was 203 nm. The linear gradient program was as follows: 0~10 min, 19% B; 10~16 min, 19~28% B; 16~30 min, 28~34% B, 30~31 min, 34~90% B; and 31~35 min, 90~90% B.

The MS analysis of ginseng samples was performed on an Agilent 6545 Q-TOF-MS system (Agilent Technologies Inc, Santa Clara, CA, USA) equipped with a Dual AJS ESI ion source. Optimized parameters for the negative-ion mode were as follows: curtain gas temperature: 320 ℃; sheath gas temperature: 320 ℃; dry gas flow rate: 8 L/min, ionization pressure: −3500 V; fragmenter: 75 V; and collision energy: 40 and 60 V. The scan mode was full scan for MS and auto scan for MS/MS. The *m*/*z* range for MS was from 400 to 1700 Da, and the *m*/*z* range for MS/MS was from 100 to 1700 Da.

### 3.5. Data Processing and Analysis

The UHPLC-Q-TOF-MS raw data from 31 batch samples and QC were analyzed using the target/suspect compound screening algorithm in the MassHunter workstation (version 10.0, Agilent Technologies Inc., Santa Clara, CA, USA). The target/suspect compound screening algorithm took all ions into account exceeding 1000 counts with a charge state equal to one, and the qualitative score of compounds was greater than 60. Isotope grouping was based on the common organic molecules model. The resulting feature for each sample screened by the workstation was exported for peak matching, aligning, and filtering. Furthermore, peaks that were lacking in more than 80% samples were removed in order to obtain common peaks. In addition, the characterization of common peaks was completed according to the formula, and the exact molecular weight and fragment refer to our existing database. The common peaks identified as ginsenosides are called characteristic ginsenosides. The peak areas of characteristic ginsenosides in all samples were used as the data matrix for subsequent data analysis, including normalization, PCA, PLS-DA, and SVM.

#### 3.5.1. Normalization Methods

The normalization methods of raw data are the mean normalization and Z-Score normalization method, whose formulas are shown below:

Mean Normalization:(1)$$P_{m,standlize}=\frac{P_{k,m}}{{\bar{P}}_{m}}$$
where $P_{m,standlize}$ refers to the peaks m in sample *k* after being normalized and ${\bar{P}}_{m}$ is the average value of peak m in all samples.

Z-Score Normalization:(2)$$P_{m,standlize}=\frac{P_{k,m}-{\bar{P}}_{k}}{\sigma_{k}}$$
where $P_{m,standlize}$ is the peaks *k* in sample m after being normalized, ${\bar{P}}_{k}$ is the average value of peak *k* in all samples, and $\sigma_{k}$ is the standard deviation of peak *k* in all samples.

#### 3.5.2. PCA Algorithm

PCA is a method of calculating principal components by covariance and using them to linearly transform the data, generally using only the first few principal components and ignoring the others [25]. The equation of the PCA model is:(3)$$\begin{array}{c} cov\left(PX\right)=E\left[PX{\left(PX\right)}^{*}\right]\\ =E\left(PXX^{*}P^{*}\right)\\ =PE\left(XX^{*}\right)P^{*}\\ =Pcov\left(XX^{*}\right)P^{-1} \end{array}$$
where *X* is the matrix of independent variables, *P* is the transformation matrix, and *PX* is a diagonal covariance matrix.

#### 3.5.3. PLS-DA Algorithm

PLS-DA is a statistical method with principal component regression. It finds a regression model by projecting the independent variable *X* and the dependent variable *Y* into a new space. PLS-DA is a variant used when *Y* is categorical [26]:

The equation of PLS model is [27]:(4)$$X=OP^{T}+E$$
(5)$$Y=UQ^{T}+F$$
where *X* is the matrix of independent variables and Y is the matrix of dependent variables; *T* and *U* are the projection of *X* and the projection vector of *Y*, respectively; *P* and *Q* are the orthogonal loading matrices; and the matrices *E* and *F* are the error terms, which are assumed to be independent and identically distributed random normal variables. The decomposition of *X* and *Y* is performed to maximize the covariance between *O* and *U*.

#### 3.5.4. SVM Algorithm

The support vector machine (SVM) model uses support vectors to learn on samples and process unknown samples with the following mathematical expression.
(6)$$w^{*}=\sum_{i=1}^{N}\alpha_{i}^{*}y_{i}x_{i}$$
(7)$$b^{*}=y_{j}-\sum_{i=1}^{N}\alpha_{i}^{*}y_{i}\left(x_{i}\cdot x_{j}\right)$$
where $\alpha_{i}^{*}$ is the constraint set for sample *i* at each iteration, $x_{i}$ is the vector composed of peak area data of sample i, $y_{i}$ is the sample label, $w^{*}$ is the feature matrix calculated at each iteration, $x_{i}\cdot x_{j}$ is the vector composed of peak area data of support vector sample *j*, $y_{i}$ is the support vector sample *j* label, and *b** is the constant vector calculated at each iteration.

The final iterative result makes sample *j* in the support vector satisfy the formula:(8)$$y_{j}\left(w^{T}x_{j}+b\right)=1$$

#### 3.5.5. Permutation Importance Algorithm

Traditional statistical learning is poorly interpretable, and calculating the feature contribution is a common method to account for sample variability. The feature contribution degree formula is calculated as follows:(9)$$imp_{k}=s-\sum_{n=1}^{N}s_{k,n}$$

All the calculation and pre-processing involving multi-model statistical analysis were performed using the Python^®^ (Version 3.7.3). SVM model and feature selection method were built by the Scikit-learn^®^ (Version 0.21.2). All raw data files were imported into python by Pandas^®^ (Version 0.25.0). 

## 4. Conclusions

In this paper, a rapid and efficient strategy is provided to achieve an intelligent distinction between ginseng from JL, LN, and HLJ. Firstly, a robust UHPLC-QTOF/MS analysis method was developed, and a total of 69 characteristic ginsenosides were successfully extracted in 31 batches of samples for subsequent analysis. PCA and PLS-DA methods could not solve the problem of the differentiation of ginseng origins, but our optimized SVM could achieve accurate differentiation, with an accuracy of 100%. More importantly, the permutation importance algorithm was used to extract quality markers in SVM for the first time, which greatly improves SVM’s interpretation ability. Finally, the test samples were accurately predicted based on the six ginsenosides coupled with SVM. The proposed approach was helpful in elaborating more the specific discrimination and prediction of ginseng and provides a simple and reliable method for the discovery of quality markers for other TCMs. 

## Figures and Tables

**Figure 1 molecules-27-04225-f001:**
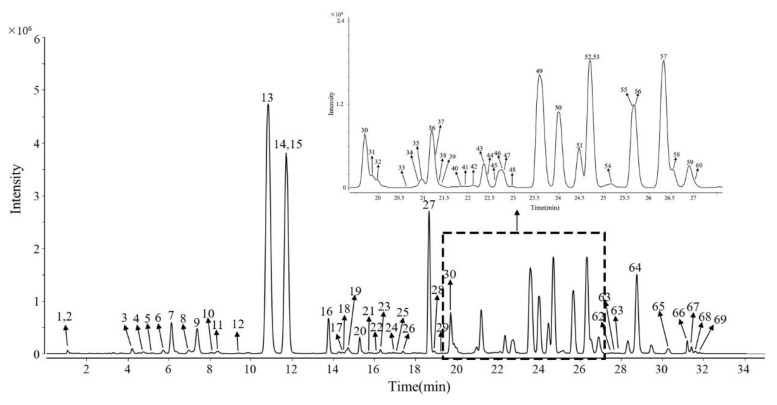
EICs of 69 characteristic ginsenosides of ginseng.

**Figure 2 molecules-27-04225-f002:**
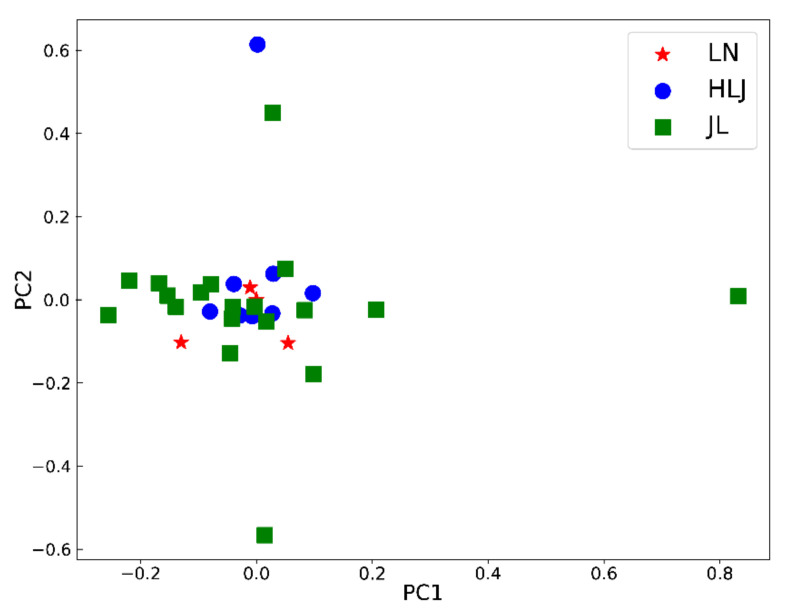
PCA results of ginseng from three geographical origins.

**Figure 3 molecules-27-04225-f003:**
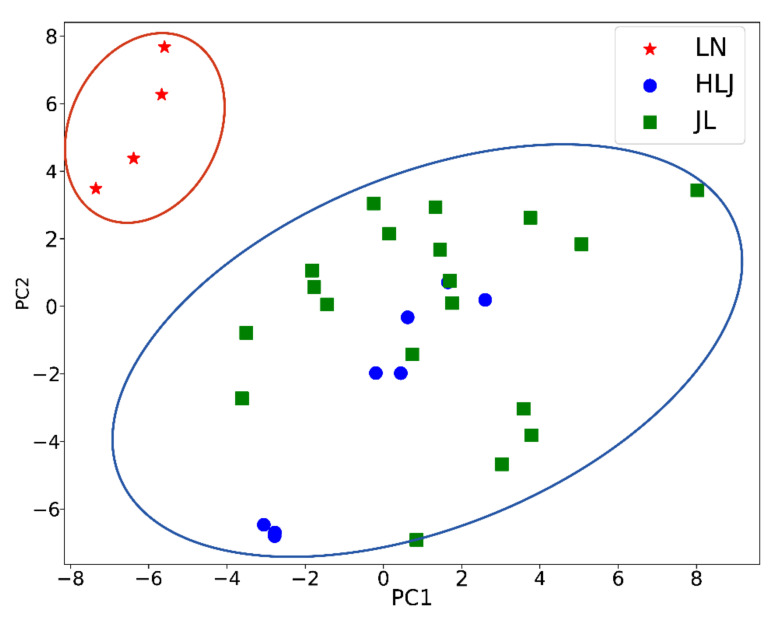
The PLS-DA results of ginsengs from three geographical origins.

**Figure 4 molecules-27-04225-f004:**
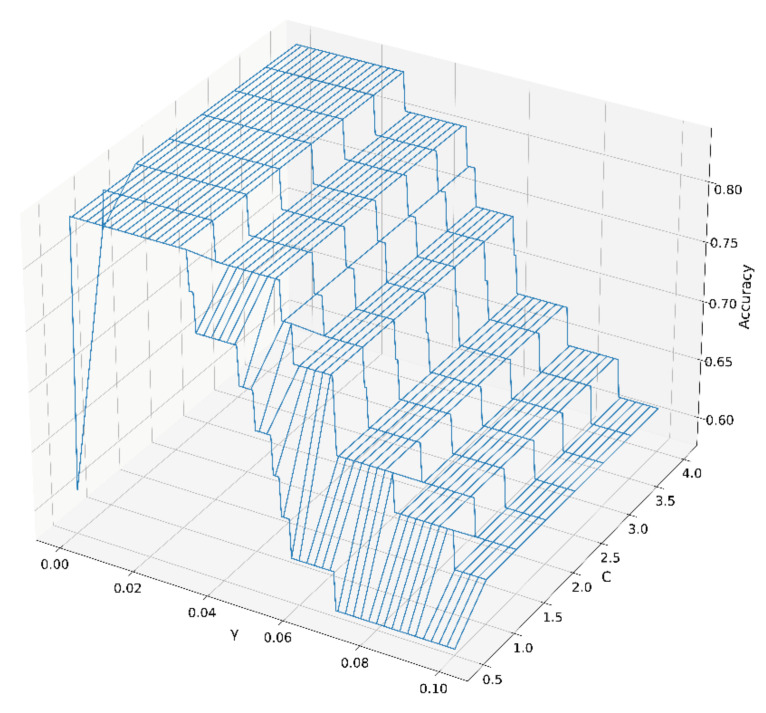
The accuracy of different parameters C and γ of the SVM model.

**Figure 5 molecules-27-04225-f005:**
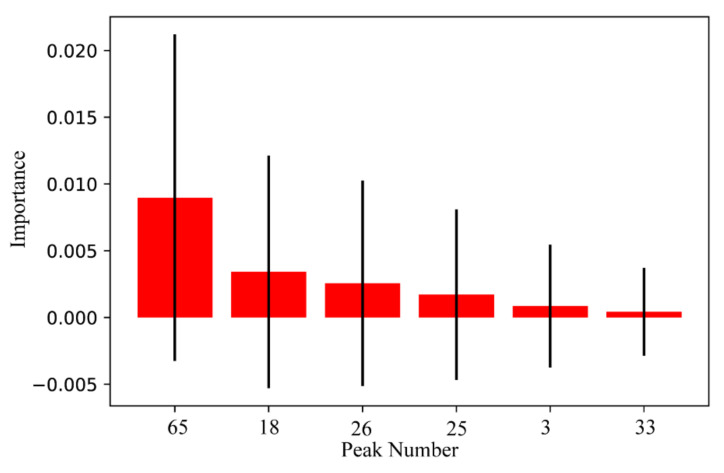
Six quality markers selected by the permutation importance algorithm.

**Table 1 molecules-27-04225-t001:** Identification of characteristic ginsenosides in ginseng by UHPLC-Q-TOF-MS.

No.	RT (min)	Compound	Formula	Mass Ion (*m*/*z*)	Type of Ion	Error (ppm)	Fragment Ions	References
1	1.07	Quinquenoside L9 or its isomer	C_42_H_74_O_15_	863.4940	[M+HCOO]^-^	−6.80	/	[22]
2	1.07	Ginsenoside Re2 or its isomer	C_48_H_82_O_19_	1007.5422	[M+HCOO]^-^	0.05	/	[16]
3	4.23	Ginsenoside Re2 or its isomer	C_48_H_82_O_19_	1007.5422	[M+HCOO]^-^	0.05	961.5401; 799.4824; 781.4713	[16]
4	4.74	(B4-b)-glc-xyl	C_41_H_70_O_14_	831.4737	[M+HCOO]^-^	−0.01	785.4677; 653.4273; 491.3746	[22]
5	5.18	Notoginsenoside R8 or its isomer	C_36_H_62_O_10_	699.4315	[M+HCOO]^-^	0.13	/	[22]
6	5.73	Ginsenoside Re4 or its isomer	C_47_H_80_O_18_	977.5353	[M-H]^-^	3.79	977.5353; 931.5271; 637.4358; 457.3784	[23]
7	6.13	Ginsenoside Re2 or its isomer	C_48_H_82_O_19_	1007.5438	[M+HCOO]^-^	1.71	961.5419; 799.4876; 637.4357; 475.3806	[16]
8	6.97	Notoginsenoside R1	C_47_H_80_O_18_	931.5260	[M-H]^-^	−1.28	931.5220; 799.4836; 638.4292; 475.3696	[23]
9	7.37	Ginsenoside Re4	C_47_H_80_O_18_	977.5330	[M+HCOO]^-^	1.43	931.5302; 637.4335; 475.3784	[23]
10	8.11	Ginsenoside Rc or its isomer	C_53_H_90_O_22_	1077.5829	[M-H]^-^	−2.06	945.5438; 719.3460; 433.5658	[21]
11	8.40	Ginsenoside Re3	C_48_H_82_O_19_	961.5365	[M-H]^-^	−1.33	799.4859; 637.4300	[23]
12	9.37	Ginsenoside Re4 or its isomer	C_47_H_80_O_18_	977.5308	[M+HCOO]^-^	−0.81	931.5203; 637.4289; 475.3736	[23]
13	10.83	Ginsenoside Rg1	C_42_H_72_O_14_	845.4912	[M+HCOO]^-^	2.27	799.4852; 637.4337; 619.4215; 475.3802	[23]
14	11.70	Ginsenoside Re	C_48_H_82_O_18_	945.5426	[M-H]^-^	−0.21	799.4880; 783.4926; 637.4346; 475.3818	[23]
15	11.70	Ginsenoside Re2 or its isomer	C_48_H_82_O_19_	961.5377	[M-H]^-^	−0.11	/	[16]
16	12.95	Vinaginsenoside R13 or its isomer	C_48_H_84_O_20_	979.5454	[M-H]^-^	−2.94	/	[22]
17	14.51	Vinaginsenoside R13 or its isomer	C_48_H_84_O_21_	979.5470	[M-H]^-^	−1.38	/	[22]
18	14.59	AcO-ginsenoside Re or its isomer	C_50_H_84_O_19_	987.5527	[M-H]^-^	−0.74	945.5519; 927.5335; 783.4923; 765.5022; 637.4373	[23]
19	14.71	AcO-ginsenoside Rf or its isomer	C_44_H_74_O_15_	841.4946	[M-H]^-^	−1.12	637.4308; 619.4205; 475.3759	[22]
20	15.43	Notoginsenoside G or its isomer	C_48_H_80_O_19_	1005.5209	[M+HCOO]^-^	−5.57	/	[22]
21	15.71	Notoginsenoside R2	C_41_H_70_O_13_	815.4791	[M+HCOO]^-^	0.31	/	[23]
22	16.29	Ginsenoside F5	C_41_H_70_O_13_	815.4789	[M+HCOO]^-^	0.16	/	[22]
23	16.19	Notoginsenoside C or its isomer	C_54_H_92_O_25_	1139.5831	[M-H]^-^	−2.08	961.5606; 785.8238; 584.0663	[22]
24	17.06	Notoginsenoside M or its isomer	C_42_H_70_O_14_	843.4734	[M+HCOO]^-^	−0.40	/	[22]
25	16.92	Ginsenoside Re2 or its isomer	C_48_H_82_O_19_	1007.5416	[M+HCOO]^-^	−0.51	961.5419; 799.4876; 637.4357; 475.3806	[16]
26	17.39	Ginsenoside Re2 or its isomer	C_48_H_82_O_19_	1007.5419	[M+HCOO]^-^	−0.20	961.5314; 799.4734	[16]
27	18.65	Ginsenoside Rf	C_42_H_72_O_14_	799.4858	[M-H]^-^	1.14	637.4327; 475.3796	[21]
28	18.90	Ginsenoside Re6 or its isomer	C_46_H_76_O_15_	913.5158	[M+HCOO]^-^	0.27	830.6457; 765.8931; 620.4240; 475.3751	[22]
29	19.24	Notoginsenoside D or its isomer	C_64_H_108_O_31_	1371.6754	[M-H]^-^	−3.49	1273.1482; 1031.7337; 875.6615; 597.4910; 415.6329	[22]
30	19.62	Notoginsenoside D or its isomer	C_64_H_108_O_31_	1371.6777	[M-H]^-^	−1.84	/	[22]
31	19.88	AcO-ginsenoside Rg1	C_44_H_74_O_15_	841.4953	[M-H]^-^	−0.26	799.4865; 679.4467; 637.4326; 619.4224; 571.3972; 475.3799	[23]
32	20.00	Notoginsenoside R4 or its isomer	C_59_H_100_O_27_	1239.6365	[M-H]^-^	−1.15	1107.5904; 1077.5822; 946.5432; 945.5391; 783.4854; 621.4298; 459.3820	[23]
33	20.53	Yesanchinoside J or its isomer	C_61_H_102_O_28_	1281.6480	[M-H]^-^	−0.37	/	[22]
34	20.98	20(R)-Ginsenoside Rh1	C_36_H_62_O_9_	683.4372	[M+HCOO]^-^	1.08	475.3815	[24]
35	20.90	Quinquenoside V	C_60_H_102_O_28_	1269.6463	[M-H]^-^	−1.76	1107.6007	[22]
36	21.20	20(R)-Ginsenoside Rg2	C_42_H_72_O_13_	829.4967	[M+HCOO]^-^	2.74	783.4923; 637.4372; 619.4248; 475.3808	[24]
37	21.33	Ginsenoside Rg5 or its isomer	C_42_H_70_O_12_	811.4842	[M+HCOO]^-^	0.54	/	[22]
38	21.36	Notoginsenoside D or its isomer	C_64_H_108_O_31_	1371.6762	[M-H]^-^	−2.95	1145.2550; 838.4987; 652.4940; 438.2765	[22]
39	21.41	Quinquenoside L1 or its isomer	C_48_H_80_O_18_	989.5313	[M+HCOO]^-^	−0.29	/	[22]
40	21.80	Ginsenoside Ra1/Ra2 or its isomer	C_58_H_98_O_26_	1209.6252	[M-H]^-^	−1.83	1077.5829; 945.5368; 783.4866; 621.7380	[23]
41	21.96	Notoginsenoside R4 or its isomer	C_59_H_100_O_2_7	1239.6356	[M-H]^-^	−1.85	1077.5843; 916.9001; 621.4288	[23]
42	22.08	Quinquenoside I or its isomer	C_52_H_86_O_19_	1059.5727	[M+HCOO]^-^	−0.63	/	[22]
43	22.11	Ginsenoside Ro or its isomer	C_48_H_76_O_19_	955.4896	[M-H]^-^	−1.29	/	[23]
44	22.35	Ginsenoside Ra1/Ra2 or its isomer	C_58_H_98_O_26_	1209.6274	[M-H]^-^	0.03	1077.5874; 945.5440; 783.4925; 621.4390;	[23]
45	22.55	Ginsenoside F1 or its isomer	C_36_H_62_O_9_	683.4380	[M+HCOO]^-^	2.25	/	[22]
46	22.70	Ginsenoside Rb1	C_54_H_92_O_23_	1153.6013	[M+HCOO]^-^	1.00	1107.5980; 945.5438; 783.4904; 621.4370; 323.0986	[21]
47	22.78	Notoginsenoside R4 or its isomer	C_59_H_100_O_27_	1239.6366	[M-H]^-^	−1.09	1209.6298; 1077.5874; 945.5440; 783.4928; 621.4390	[23]
48	22.98	Ginsenoside Ra1/Ra2 or its isomer	C_58_H_98_O_26_	1209.6188	[M-H]^-^	−7.10	1077.5826; 945.5420	[23]
49	23.58	Ginsenoside Ro	C_48_H_76_O_19_	955.4924	[M-H]^-^	1.63	793.4392; 569.3860; 455.3534;	[23]
50	24.01	Ginsenoside Rc	C_53_H_90_O_22_	1123.5915	[M+HCOO]^-^	1.77	1077.5879; 945.5412; 915.5334	[21]
51	24.46	Ginsenoside Ra1/Ra2 or its isomer	C_58_H_98_O_26_	1209.6282	[M-H]^-^	0.68	1077.5846; 945.5437; 915.5327; 783.4863	[23]
52	25.04	Ginsenoside F1 or its isomer	C_36_H_62_O_9_	683.4371	[M+HCOO]^-^	0.94	/	[22]
53	25.04	AcO-ginsenoside Ro	C_50_H_78_O_20_	997.5001	[M-H]^-^	−1.30	/	[22]
54	25.19	Ginsenoside Ra1/Ra2 or its isomer	C_58_H_98_O_26_	1209.6254	[M-H]^-^	−1.69	1077.5842; 783.4910; 621.4377	[23]
55	25.62	Ginsenoside Ra1/Ra2 or its isomer	C_58_H_98_O_26_	1209.6226	[M-H]^-^	−3.94	1077.5856; 621.3146	[23]
56	25.67	Ginsenoside Rb2	C_53_H_90_O_22_	1123.5918	[M+HCOO]^-^	2.04	1077.5824; 945.5402; 915.5279; 783.4881; 765.4772; 621.4359;	[21]
57	26.24	Ginsenoside Rb3	C_53_H_90_O_22_	1123.5902	[M+HCOO]^-^	0.66	1077.5892; 945.5474; 915.5364; 783.4912; 621.4374; 459.3830	[23]
58	26.77	Quinquenoside L1 or its isomer	C_48_H_80_O_18_	943.5262	[M-H]^-^	−1.04	/	[22]
59	26.90	m-Ginsenoside Rc/Rb2 or m-Ginsenoside Rb3	C_56_H_92_O_25_	1163.5858	[M-H]^-^	0.23	1119.6012; 1077.5910; 1059.5793; 915.5332; 765.4795	[23]
60	27.00	Ginsenoside Ra1/Ra2 or its isomer	C_58_H_98_O_26_	1209.6257	[M-H]^-^	−1.38	/	[23]
61	27.37	Notoginsenoside O or its isomer	C_52_H_88_O_21_	1093.5787	[M+HCOO]^-^	−0.15	/	[22]
62	27.57	Yesanchinoside J or its isomer	C_61_H_102_O_28_	1281.6451	[M-H]^-^	−2.62	/	[22]
63	27.87	Vinaginsenoside R3 or its isomer	C_48_H_82_O_17_	975.5511	[M+HCOO]^-^	−1.22	739.7635; 576.8463; 481.3275; 324.4059	[22]
64	28.76	Ginsenoside Rd	C_48_H_82_O_18_	991.5497	[M+HCOO]^-^	2.55	945.5477; 783.4920; 765.480; 621.4385; 459.3882	[21]
65	30.28	AcO-ginsenoside Rd or its isomer	C_50_H_84_O_19_	987.5526	[M-H]^-^	−0.84	987.5518; 945.5420; 927.5342; 783.4925; 765.4773; 621.4397; 459.3808	[23]
66	31.41	Quinquenoside L14 or its isomer	C_47_H_80_O_17_	961.5370	[M+HCOO]^-^	0.27	915.5347; 783.4907; 709.1200; 621.4368; 434.0248	[22]
67	31.43	Ginsenoside Re2 or its isomer	C_48_H_82_O_19_	961.5403	[M-H]^-^	2.60	/	[16]
68	31.55	Quinquenoside I or its isomer	C_52_H_86_O_19_	1059.5731	[M+HCOO]^-^	−0.27	915.5271; 783.4907; 621.4369; 459.3846	[22]
69	31.68	Quinquenoside I or its isomer	C_52_H_86_O_19_	1059.5852	[M+HCOO]^-^	11.10	/	[22]

**Table 2 molecules-27-04225-t002:** The classified and predicted results of ginsengs from three geographical origins using SVM model with raw data and normalized data.

Raw Data (Accuracy = 83%)						Normalized Data (Accuracy = 100%)					
Sample	Actual	Recognized	Sample	Actual	Recognized	Sample	Actual	Recognized	Sample	Actual	Recognized
S1	LN	JL	S17	JL	JL	S1	LN	LN	S17	JL	JL
S2	LN	LN	S18	JL	JL	S2	LN	LN	S18	JL	JL
S3	LN	JL	S19	JL	JL	S3	LN	LN	S19	JL	JL
S4	LN	JL	S20	JL	JL	S4	LN	LN	S20	JL	JL
S5	HLJ	HLJ	S21	JL	JL	S5	HLJ	HLJ	S21	JL	JL
S6	HLJ	HLJ	S22	JL	JL	S6	HLJ	HLJ	S22	JL	JL
S7	HLJ	HLJ	S23	JL	JL	S7	HLJ	HLJ	S23	JL	JL
S8	HLJ	JL	S24	JL	JL	S8	HLJ	HLJ	S24	JL	JL
S9	JL	JL	S25	JL	JL	S9	JL	JL	S25	JL	JL
S10	JL	JL	S26	JL	JL	S10	JL	JL	S26	JL	JL
S11	JL	JL	S27	JL	JL	S11	JL	JL	S27	JL	JL
S12	JL	JL	S28	JL	JL	S12	JL	JL	S28	JL	JL
S13	HLJ	HLJ	S29	JL	JL	S13	HLJ	HLJ	S29	JL	JL
S14	HLJ	HLJ	S30	JL	JL	S14	HLJ	HLJ	S30	JL	JL
S15	HLJ	JL	S31	JL	JL	S15	HLJ	HLJ	S31	JL	JL
S16	HLJ	HLJ				S16	HLJ	HLJ			

**Table 3 molecules-27-04225-t003:** The classified and predicted results of ginseng test samples from three different geographical origins using SVM model with raw data and normalized data.

Sample	Actual	Recognized
S32	LN	LN
S33	HLJ	HLJ
S34	JL	JL
S35	JL	JL
S36	JL	JL
S37	JL	JL
S38	JL	JL
S39	JL	JL

**Table 4 molecules-27-04225-t004:** Sample information of 39 batches of ginseng *.

No.	Origin	Age	Batch Code	No.	Origin	Age	Batch Code
S1	Dandong City, Liaoning Province	4	20200901	S21	Changbai County, Jilin Province	5	20190901
S2	Dandong City, Liaoning Province	4	20200902	S22	Changbai County, Jilin Province	5	20190902
S3	Dandong City, Liaoning Province	4	20200903	S23	Changbai County, Jilin Province	5	20190903
S4	Dandong City, Liaoning Province	4	20200904	S24	Changbai County, Jilin Province	5	20190904
S5	Mudanjiang City, Heilongjiang Province	5	RS180321-2	S25	Ji’an City, Jilin Province	5	20180421-1
S6	Mudanjiang City, Heilongjiang Province	5	RS180322-2	S26	Ji’an City, Jilin Province	5	20180421-2
S7	Mudanjiang City, Heilongjiang Province	5	RS180323-2	S27	Ji’an City, Jilin Province	5	20180421-3
S8	Mudanjiang City, Heilongjiang Province	5	RS180324-2	S28	Ji’an City, Jilin Province	5	20180421-4
S9	Tonghua City, Jilin Province	5	RS180311	S29	Fusong County, Jilin Province	5	20180911-1
S10	Tonghua City, Jilin Province	5	RS180312	S30	Fusong County, Jilin Province	5	20180911-3
S11	Tonghua City, Jilin Province	5	RS180313	S31	Fusong County, Jilin Province	5	20180911-4
S12	Tonghua City, Jilin Province	5	RS180314	T1	Liaoning Province	/	/
S13	Heilongjiang Province	5	RS180321	T2	Heilongjiang Province	/	/
S145	Heilongjiang Province	5	RS180322	T3	Jilin Province	/	/
S15	Heilongjiang Province	5	RS180323	T4	Heilongjiang Province	/	/
S16	Heilongjiang Province	5	RS180324	T5	Jilin Province	/	/
S17	Jingyu County, Jilin Province	5	20190901	T6	Jilin Province	/	/
S18	Jingyu County, Jilin Province	5	20190902	T7	Jilin Province	/	/
S19	Jingyu County, Jilin Province	5	20190903	T8	Jilin Province	/	/
S20	Jingyu County, Jilin Province	5	20190904				

***** All samples were cultivated, and roots were used in the experiment.

## Data Availability

The authors confirm that the data supporting the findings of this study are available within the article and from the corresponding author upon request.

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
