# Peer review of "Rapid Discrimination and Prediction of Ginsengs from Three Origins Based on UHPLC-Q-TOF-MS Combined with SVM"

_molecules, 2022, doi:10.3390/molecules27134225_

Round 1

Reviewer 1 Report

Molecules       Manuscript ID:  molecules-1757989 

Title:  Rapid discrimination and predication of ginsengs in three ori-gins based on UHPLC-Q-TOF-MS combined with SVM

Authors Chi Zhang, Zhe Liu, Shaoming Lu, Liujun Xiao, Qianqian Xue, Hongli Jin, Jiapan Gan, Xiaonong Li, Yan-fang Liu and Xinmiao Liang

In this article, the authors discuss an important topic about the quality and characteristics of ginseng of various origins. Using ultra-high performance liquid chromatography quadrupole mass spectrometry (UHPLC-Q-TOF-MS) with a support vector machine (SVM), the authors developed a stable and reliable new method for rapidly distinguishing and predicting ginseng from three main regions of cultivation in China. A method for rapid pretreatment, for rapid screening and identification of 69 characteristic ginsenosides in 31 batches of ginseng samples from three different origins is presented.

The first new SVM-optimized method achieves 100% differentiation of different ginseng extracts using state-of-the-art and accurate equipment such as UHPLC-Q-TOF-MS.

More than 6000 pieces of information from 31 batch samples were analyzed, compared to a database (including MS and MS / MS data for over 400 ginsenosides collected from published references), 69 ginsenosides were quickly tested and their chemical structures were identified in advance.

The proposed strategy and new method are suitable for discrimination and prediction of ginseng origin. Published results provide a simple and reliable method for finding quality tags for other TCMs.

Author Response

We want to begin by thanking the reviewer 1 for writing that “The proposed strategy and new method are suitable for discrimination and prediction of ginseng origin. Published results provide a simple and reliable method for finding quality tags for other TCMs.” We have added and improved the content of the manuscript to make it more complete.

Reviewer 2 Report

Section 2.1.1. authors claim that in order to achieve good separation effect and obtain high-quality UHPLC-Q-TOFMS data, they have optimized the extraction method, extraction solvents, composition

Of mobile phase, elution gradient and injection concentration in detail during UHPLC-Q-TOF-MS” They haven’t provide any data to support this statement. And accutally they should be added to the paper. So please, before publication, include the data on sample extraction conditions optimization, provide data for the optimization of mobile phase, stationary phase, elution conditions. Include all of these data in the main article or in the supplementary materials. Figure 1S provides only partial information, it is only chromatograms (it is not enough) in addition these chromatograms are not described and in case of solvent selection for extraction I do not know which chromatogram represents which solvent and in general which solvents were tested. Moreover, if such results are presented in the supplement, they should be discussed in detail in the main text and not mentioned in two sentences: The injection concentration of 2000 ppm can obtain excellent response

and will not burden the instrument. Those results are shown in Figure S1. Those results

show that optimization of UHPLC-Q-TOF-MS analysis conditions of ginseng is very necessary to ensure that the samples enter the subsequent analysis in the best state.

Section 2.1.2. validation involves usually the determination of calibration curve, LOD, LOQ, inter-day, intra day precision. What about these data?

Figure 1 is for standards mixture, or extract? For which ion is this EIC chromatogram? EIC is usually performed for one m/z value, for one ion. Are all of these peaks coming from one ion, one compound? Provide the notations for each peak. As far as I can see compounds 1 and 2 are eluted in the dead volume, you should change the gradient program, or exclude these two compounds from your results.

Can you please include TIC chromatograms of your extracts in the main text or in supplement? I suppose you will have many other peaks, because the extraction method is not selective at all, therefore it should be clearly indicated in the text. In this way, other authors will be sure that the use of your method will not only allow the selective extraction of this group of compounds but also many others.

Please determine the matrix effect and include this data in the publication. I think that with such a non-selective extraction method it will be high. It must be determined.

Will the optimized SVM method allow achieving accurate differentiation for other samples? Will it be universal for the differentiation of samples of other origins? Now the n=interest to the readers is low in my opinion, as these data concern only samples from China. If authors prove its universality (and utility), it will surely gain in value.

Reviewer 3 Report

My main objection to the reviewed manuscript ID molecules-1757989 is that even a tentative identification of the compounds should be better documented (Table 1). Especially as  some of the data given in Table 1 are disputable to say at least. I can understand that for the low abundant compounds there is a lack of the product ions (e.g. 1, 2, 5, 15…). However, it is strange that sometimes the isomers have so very different retention times and completely different product ions. Namely, ginsenoside Re2 and its isomers. The first isomer has rt=1.07 and the last one has rt=31.43. The isomer at rt=16.92 has product ions at m/z 815.8789, 601.8789, 403.5587 and 241.2822, whereas that at rt=17.39 has product ions at m/z 961.5314, 799.4734, 602.5592, 368.8669, 179.8671 (some the values, e.g. 368.8669, also seem to be  strange). Therefore, I strongly suggest the authors  to carefully analyze their whole UHPLC-Q-TOF-MS data once again. At least for a few compounds (e.g. at least for five) the spectra obtained in the full scan mode and MS/MS mode should be shown (e.g. in the supplementary material). Published data concerning fragmentation of at least a few exemplary compounds should be briefly discussed and compared to the obtained MS/MS data to show that the proposed identification is plausible. Because of the above, I suggest a major revision and re-review (although I must admit I considered suggesting rejection), hopefully, the UHPLC-Q-TOF-MS data in the revised manuscript will be reliable. Of course, another problem is how the revision of the UHPLC-Q-TOF-MS data will affect the results of SVM analysis and others. It should be explained in the authors’ reply.

Other comments are as follow:

1.     Almost everywhere in the manuscript, the space between cited reference and preceding word should be added (e.g. page 1 -  effects[1].).

2.     Almost in each figure the fonts on the axis should be larger, especially Figure S1 and Figure 4.

3.     Page 2 – why PCA and PLS-DA “…lacked objectivity and accuracy in identification results.”?

4.     Page 2 – abbreviation TCM should be explained.

5.     Page 2 – “girpmseng”?

6.     Page 2 – “The injection concentration of 2000 ppm can obtain excellent response and will not burden the instrument. Those results are shown in Figure S1.” However, in Figure S1 there are no results for the concentration 2000 ppm. The data of extraction solvents used for the left chromatograms in Figure S1, if I understand correctly the caption, should be provided.

7.     Table 1, 6th column, instead of “Ion mode it would be better "Type of ion" and ion assignment as [M+HCOO]- and [M-H]-.

8.     Page 9, subsection 3.1, it should be “Table 4”

9.     Page 10, “…and the detection wavelength was 203 nm”. Were the LC-UV data also analyzed in the work?

Round 2

Reviewer 2 Report

Authors have implemented all corrections and have answered all reviewer questions. The paper may be accepted for publication.

Author Response

Thank you very much for your insightful advice, we could not complete this work without your assistance.

Reviewer 3 Report

For me, as reviewer, it was one of the most difficult decisions, to suggest acceptance  (minor revision) or rejection. The data presented in Table 1 (the data are of crucial importance for the rest of the manuscript) are still disputable to say the least. However, I decided to accept the fact that the data obtained by the authors were as presented. I only wish to stress that the authors, not reviewers, are fully responsible for the manuscript content. I suggest adding, in the subsection 2.2 a few references which would support the data. The formate adducts should be assigned as [M+HCOO]-, not as [M+COOH]-.
